# Neurocognitive profile in HIV subjects on INSTI-regimen- one year follow up: Is there room for optimism?

**Nina Brkić-Jovanović**[1]*, **Mina Karaman**[1], **Vanja Andrić**[2], **Daniela Marić**[2], **Snežana Brkić**[2], **Vojislava Bugarski-Ignjatović**[1]

**1** Department of Psychology, Faculty of Medicine, University of Novi Sad, Novi Sad, Serbia, **2** Department of Infectious Diseases, Faculty of Medicine, University of Novi Sad, Novi Sad, Serbia

* nina.brkic-jovanovic@mf.uns.ac.rs

**Data Availability Statement:** All relevant data are within the manuscript and its Supporting information files.

## Abstract

The introduction of antiretroviral therapy (ART) has successfully changed the clinical course of people with HIV, leading to a significant decline in the incidence of HIV-related neurocognitive disorders. Integrase strand transferase inhibitors (INSTI) are recommended and preferred first-line ART for the treatment of HIV-1 infection in ART-naïve subjects. This type of therapy regimen is expected to have higher CNS penetration, which may bring more cognitive stability or even make significant cognitive improvement in people with HIV. The study aimed to follow up on neurocognitive performance in HIV subjects on two types of INSTI therapy regimens at two-time points, one year apart. The study sample consisted of 61 ART naïve male participants, of which 32 were prescribed raltegravir (RAL) and 29 dolutegravir (DTG). There was no significant difference between subsamples according to the main sociodemographic (age, education level) and clinical characteristics (duration of therapy, nadir CD4 cells level, CD4 cells count, CD8 cells, CD4/CD8 ratio). For neurocognitive assessment, six measures were used: general cognitive ability (MoCA test), verbal fluency (total sum score for phonemic and category fluency), verbal working memory (digit span forward), cognitive capacity (digit span backwards), sustained attention (Color Trail Test 1), and divided attention (Color Trail Test 2). In both therapy groups (RAL and DTG), there was no significant decrease in neurocognitive achievement on all used measures over a one-year follow-up in both therapy groups. A statistically significant interactive effect of time and type of therapy was found on the measure of divided attention—DTG group showed slight improvement, whereas RAL group showed slight decrease in performance. During the one-year follow-up of persons on INSTI-based regimen, no significant changes in cognitive achievement were recorded, which suggests that the existing therapy can have a potentially positive effect on the maintenance of neurocognitive achievement.

## Introduction

The epidemiology of HIV infection in Serbia is similar to that in developed countries, with an average age of patients around 50 years, predominantly infected MSM population [1]. With

**Funding:** This work is supported in part by a research grant from the Investigator-Initiated Studies Program of Merck Sharp & Dohme doo.

**Competing interests:** The authors have declared that no competing interests exist.

the successful peripheral suppression of HIV after the introduction of antiretroviral therapy (ART), HIV disease has changed its course. It now represents a chronic disease, with most patients reaching senior age [2, 3]. In these patients, ageing-associated comorbidities, especially in cognitive functioning, play a central role in the overall quality of life [2].

Recent studies have shown that HIV enters the central nervous system (CNS) in the first few days after transmission and causes an acute inflammatory reaction in the brain, promptly limited by the introduction of ART [4]. There are several ways in which HIV infection is compartmentalised in the CNS. Besides the persistence of the virus in microglia cells and macrophages, parts of the virus can cause the persistence of inflammation and neurodegeneration. Thus, the virus only triggers the inflammatory response in the brain, which is afterwards maintained in the form of low-level neuroinflammation of the microglia and macrophages, resulting in continuous and diffuse neuronal death or dysfunction and leading to a certain level of neurodegeneration [3, 5]. With the ageing of people with HIV (PWH), this form of neurodegeneration is combined with the physiological ageing of the brain, most probably synergistically [6, 7]. This type of brain ageing is referred to as accentuated and is not specific to HIV infection [4].

The concept of modern treatment for HIV infection consists of the prompt introduction of ART. Irrespective of nadir CD4 or as soon possible, the combination of drugs is designed to allow successful peripheral viral suppression and perform beneficial effects on halting the damage in the CNS [8, 9]. Integrase strand transferase inhibitors (INSTI) nowadays are recommended third agents in the treatment of HIV infection [10]. Due to their favourable side effect profile, limited drug-drug interactions, and virologic potency, INSTI-based regimens are now among the recommended and preferred first-line ART for the treatment of HIV-1 infection in ART-naïve patients. Dolutegravir is a second-generation INSTI with some advantages compared to other available antiretroviral agents [11–13]. Raltegravir is the first HIV-integrase inhibitor approved by the FDA for the treatment of HIV infection [14].

INSTI are influential inside the cell by preventing the integration of proviral DNA in the cell genome. They are expected to be more successful in preserving CNS reservoirs, enabling a reduced effect in relation to the patient's neurocognitive condition [15]. Also, with the ART regimen, the incidence of HIV-associated neurocognitive disorders (HAND) has started to decline, primarily in severe forms of HAND, such as HIV-associated dementia (HAD) [16–18]. The benefits of ART can be seen in the improvement of neurocognitive performance in more than 40% of individuals with HAD [18].

However, a milder form of HAND still remains. It persists in 15–55% of people with HIV in the ART era, mainly in the form of asymptomatic neurocognitive impairment (ANI) and mild neurocognitive disorder (MND) [19–21]. Typically, HAND includes executive dysfunction and memory impairment with prominent disruption of attention, multitasking, impulse control, judgement and memory encoding and retrieval [22]. HAND can also be associated with motor dysfunction, including bradykinesia, loss of coordination and gait disturbances [22]. In the pre-ART era, dominant neurocognitive impairments were deficits in psychomotor speed, motor dysfunction such as gait disturbances, and memory impairment [23]. In the current ART era, HAND presents as a mixed pattern of cortical and subcortical deficits with most impairment in executive function, learning and working memory [24].

HAND in PWH has several possible pathogenic mechanisms, including poor drug concentration of antiretrovirals in CNS, the legacy effect of CNS damage sustained during early stages of HIV infection of the brain, antiretroviral neurotoxicity, persistent brain immune activation, and comorbidities such as cerebrovascular disease, syphilis, and hepatitis C co-infection [25–28]. Besides HIV-related factors, some sociodemographic characteristics such as older age, race, and education level [23] and depression [22] may influence the likelihood of

developing HAND [15]. Also, the type of therapy regimen can influence neurocognitive performance, where ART regimens with higher versus lower CNS penetration may affect more significant cognitive improvement in individuals with HAD [29].

HAND generally remains stable during ART but rarely resolves completely [20]. One 4-year follow-up study of ART-treated individuals demonstrated that 77% remained neurocognitively stable, with only 13% deteriorating to a more severe form of HAND and 10% improving [30]. Thus, HAND is typically not progressive in most aviraemic PWH on ART [20]. The fact that lower CD4+ T cell nadir is one of the risk factors for HAND suggests that earlier HIV treatment to prevent severe immunosuppression could reduce the severity of HAND [31]. Also, a recent study suggests that ART initiation very shortly after HIV acquisition results in a more significant improvement in PWH neurocognitive performance over time compared to deferred ART treatment 24 weeks later [17].

Based on the current knowledge, the present study aimed to follow up on neurocognitive performance in PWH on two types of ART therapy regimens (INSTI) at two-time points, one year apart.

## Method

A longitudinal, comparative and correlational research design survey was applied from September 2021 to September 2023 to a convenience sample of persons with HIV who have been on INSTI-based therapy since the beginning of treatment. Repeated measurements were carried out one year apart, and comparing the participants to the baseline observed in the first test was possible.

### Sample and setting

The study was conducted at the Clinical Center of Vojvodina, Novi Sad, Republic of Serbia. The studied sample consisted of 61 PWH who received INSTI-based treatment. The sample consisted of 61 male participants, of which 32 were prescribed raltegravir (RAL) and 29 dolutegravir (DTG) as part of the therapy for HIV. All participants were INSTI naïve, starting therapy as soon as possible after diagnosis. In order to participate in the study, participants had an undetectable viral load at least 6 months before the baseline test. The exclusion criteria were the history of neurological or psychiatric illnesses, taking any medications that are not related to HIV therapy and comorbidities known to influence cognitive performance (diabetes, hepatitis C, thyroid and cardiovascular diseases). All participants underwent a 60-minute neurocognitive test battery at two-time points (baseline and second assessments), one year apart, which tested their achievement in different neurocognitive domains.

Using analytics calculators sample size software for multiple regression, a sample size of 31 participants is required for a 95% confidence interval, with expected small effect size of 0.35 and a margin of error of 0.05, with 2 predictor variables and 0.8 statistical power. The number of people living with HIV in Vojvodina is relatively small, and with the mentioned criteria for inclusion and exclusion from the study, this number of 62 respondents represents the entire population.

### Ethics approval and consent to participate

The implementation of this study was approved by the Ethics Committee of the Faculty of Medicine, the University of Novi Sad, Serbia (01-39/60/1) and Ethics Commission of The Clinical Center of Vojvodina (00-144/6/21). The study was conducted in compliance with the Helsinki Convention principles. Participation in the study was voluntary. All participants signed a

written consent and were informed of the study's objectives. Anonymous answers guaranteed data confidentiality.

## Variables and instruments

Sociodemographic (age, education level) and clinical data (duration of therapy, nadir CD4 cells level, CD4 cells count, CD8 cells, CD4/CD8 ratio, type of INSTI (RAL/DTG) were collected from patients' medical records.

1. Neurocognitive assessment comprises several neuropsychological tests. The Montreal Cognitive Assessment test (MoCA) is a widely used screening assessment) for detecting cognitive impairment. This test consists of 30 points and takes 10 minutes for the individual to complete. The MoCA assesses several cognitive domains: short-term memory recall tasks, visuospatial abilities, attention, concentration and working memory, executive functions, language, abstract reasoning and orientation [32]. We used the MoCA general score as a continuous variable to measure overall cognitive performance for further analysis. We used the MoCA test as a screening tool for HAND among PWH, with a cutoff score of 23.

2. Verbal fluency test, which was measured using the Phonemic Fluency Test, including phonemes S/K/L in the Serbian language equivalent to the Verbal Fluency Test (FAS) in the English language [33], and the "Animals" subtest from the Boston Diagnostic Aphasia Exam [34]. In further analysis, we use the composite and standardised score for phonemic and category fluency as continuos variables.

3. Attention/Working memory was measured using the Wechsler Memory Scale-Revised [35]. We used the subtest Digit span—forward test to measure verbal working memory and attention and the Digit span—backwards test to measure cognitive capacity. We used the general score as a continuous variable to measure performance on these variables, where a higher score indicates a better achievement.

4. The Color Trails Test (CTT) is a language-free version of the Trail Making Test (TMT) that was developed to allow for broader cross-cultural application to measure sustained attention (CTT 1) and divided attention (CTT 2) in adults [36]. We used time reaction as a continuous variable to measure performance on these variables, where a higher score indicates a lower achievement.

Six measures from these tests were used in a study analysis as a measure of general cognitive ability (MoCA test), verbal fluency (composite and standardised score for phonemic and category fluency), verbal working memory (digit span forward), cognitive capacity (digit span backwards), sustained attention (time reaction on CTT 1) and divided attention (time reaction on CTT 2).

## Statistical analysis

Frequency and percentage displays were used to represent a specific category of answers to analyse and describe the sample structure by relevant variables. Statistical analysis was conducted using SPSS (IMB V.26), specifically descriptive statistics, Pearson correlations, t-tests, and MANCOVA. Given that two different scales were used to measure verbal fluency, the composite variable was formed first by converting raw scores into z-scores. The individual z-scores were then summed and divided by 2 to obtain an average composite score, which was then converted to a T-score (T (50,10)).

In the applied tests, the limit values of the risk probability are at the significance level of 95% ($p < 0.05$) (difference in statistical parameters significant) and 99% ($p < 0.01$) (difference in

statistical parameters highly significant). Preliminary analyses of the effects of the age and duration of therapy were performed to determine whether potential correlations should be considered in further statistical analysis. The strength of the linear relationship between the duration of therapy and performance in neurocognitive domains was tested using Pearson's correlation coefficient separately for two testing moments.

## Results

The Pearson coefficient was applied to examine the correlation of sociodemographic variables and backbone therapy with neurocognitive measures. Preliminary testing was done to verify the normality, linearity, univariate and multivariate outliers, homogeneity of variance and covariance, and multicollinearity of variables.

In order to verify the connection between age, education and duration of therapy with neurocognitive measures, correlation coefficients were conducted. The participants were selected to all have at least a high school education, an average of 13.43 educational years. The age in the sample varied from 22 to 68, with an average age of 38.32 years. Duration of the therapy for HIV varied from 21 to 71 months, with an average of 41 months of treatment.

Since back bone therapy can affect neurocognitive achievements, we used and checked this as a controlling variable. Three types of backbone therapy were represented: tenofovir (40, 66.7%), abacavir (13, 21.7%) and lamivudine (7, 11.7%).

Bivariate correlation tests showed no significant correlations between age, therapy duration, education level, backbone therapy and neurocognitive measures (Table 1).

To check the potential effect of infection progression through nadir CD4 cells (nCD4) as an indicator in two groups of PWH on two types of therapy, we conducted a t-test for independent samples. The results show that the subjects on RAL do not differ significantly from the subjects on DTG in the number of nCD4 cells, and we can consider them equal in terms of that parameter (t(58) = 1.171, p = 0.246).

The average scores on every measure, dependent on the therapy prescribed and the time of assessment, are shown in Table 2.

In order to check the representation of HAND in the sample at the baseline assessment and in the second assessment, we used the MoCA test because recent studies indicate that it is a "reasonable screening tool" for HAND among PWH, with a cutoff score of 23 as an optimal balance in relation to sensitivity and specificity [37]. According to this criterion, in the baseline assessment, 8 participants met the criteria for HAND. In the second assessment, there were 3 participants with HAND, which shows that although there are no statistically significant differences in the score on the MoCA test, there is still a slight improvement.

To test significant changes in the neurocognitive achievement of PWH during INSTI therapy of DTG and RAL, MANCOVA repeated measure was conducted, controlling the duration of the therapy, for every neurocognitive measure: MOCA, verbal working memory, cognitive capacity, sustained attention, divided attention, and verbal fluency. Additionally, changes in CD4 cells and CD8 cells were also tested. A statistically significant interactive effect of time

**Table 1. Bivariate correlation between age, education level, duration of the therapy and neurocognitive measures.**

| Neurocognitive measures | MOCA | Verbal Fluency | Verbal working memory | Cognitive capacity | Sustained attention | Divided attention |
|---|---|---|---|---|---|---|
| Age | -.233 | -.190 | -.150 | -.135 | .246 | .097 |
| Therapy duration | -.087 | -.065 | -.106 | -.052 | .155 | .236 |
| Education level | .211 | .128 | .072 | .081 | .147 | .248 |
| Backbone therapy | .128 | .165 | .052 | .120 | -.076 | -.098 |

**Table 2. Descriptive statistics based on type of therapy and time of assessment.**

| Neurocognitive measures | Therapy | Baseline assessment | | Second assessment | |
|---|---|---|---|---|---|
| | | Mean | SD | Mean | SD |
| MOCA | Raltegravir | 25.39 | 3.64 | 26.20 | 2.56 |
| | Dolutegravir | 26.08 | 2.56 | 26.30 | 2.00 |
| Verbal Fluency | Raltegravir | 49.17 | 16.83 | 47.21 | 13.66 |
| | Dolutegravir | 50.90 | 16.43 | 52.08 | 20.18 |
| Verbal working memory | Raltegravir | 6.75 | 1.10 | 6.60 | 0.91 |
| | Dolutegravir | 6.58 | 1.15 | 6.95 | 0.75 |
| Cognitive capacity | Raltegravir | 4.62 | 1.33 | 5.00 | 1.25 |
| | Dolutegravir | 4.51 | 0.98 | 4.50 | 1.10 |
| Sustained attention | Raltegravir | 43.51 | 15.95 | 32.35 | 10.55 |
| | Dolutegravir | 36.20 | 12.84 | 37.10 | 12.00 |
| Divided attention | Raltegravir | 87.83 | 30.33 | 80.13 | 25.79 |
| | Dolutegravir | 69.37 | 28.92 | 83.60 | 23.47 |
| CD4 cells | Raltegravir | 1182.74 | 490.08 | 1039.43 | 426.14 |
| | Dolutegravir | 998.75 | 469.96 | 995.44 | 424.65 |
| CD8 cells | Raltegravir | 1369.22 | 634.32 | 1184.14 | 475.44 |
| | Dolutegravir | 1471.79 | 519.50 | 1415.00 | 581.09 |
| CD4/CD8 ratio | Raltegravir | 1.33 | 2.25 | 1.03 | 0.27 |
| | Dolutegravir | 1.07 | 1.87 | 0.77 | 0.39 |

and type of therapy was found on a measure of divided attention (F (6, 71) = 9.16, p = .005; partial eta squared = .223) (Table 3).

Fig 1 indicates that DTG group showed slight improvement on the divided attention measure, whereas the RAL group showed slight decrease in performance.

**Table 3. MANCOVA's main and interactive effect results.**

| Test | Effect | F | p | Partial eta |
|---|---|---|---|---|
| MOCA | Time | .119 | .733 | .004 |
| | Time * type of therapy | .319 | .576 | .010 |
| Verbal Fluency | Time | .450 | .507 | .014 |
| | Time * type of therapy | .970 | .332 | .029 |
| Verbal working memory | Time | .333 | .568 | .010 |
| | Time * type of therapy | 1.212 | .279 | .036 |
| Cognitive capacity | Time | .881 | .355 | .027 |
| | Time * type of therapy | 2.047 | .162 | .060 |
| Sustained attention | Time | .750 | .393 | .023 |
| | Time * type of therapy | .239 | .628 | .007 |
| Divided attention | Time | 1.163 | .289 | .035 |
| | Time * type of therapy | 9.164 | .005 | .223 |
| CD4 cells | Time | 3.145 | .082 | .058 |
| | Time * type of therapy | 2.317 | .207 | .031 |
| CD8 cells | Time | .062 | .804 | .001 |
| | Time * type of therapy | .349 | .557 | .007 |
| CD4/CD8 ratio | Time | .037 | .848 | .001 |
| | Time * type of therapy | .153 | .698 | .004 |

Scheffe's post-hoc test yielded no significant differences in individual variable contributions (Table 4).

**Table 4. Scheffe post-hoc.**

| Variable A | Variable B | Mean difference (A-B) | SE | t | p |
|---|---|---|---|---|---|
| Raltegravir, 1 | Dolutegravir, 1 | 18.13 | 9.944 | 1.82 | 0.35 |
|  | Raltegravir, 2 | 12.89 | 6.740 | 1.91 | 0.31 |
|  | Dolutegravir, 2 | 3.65 | 9.88 | 0.37 | 0.98 |
| Dolutegravir, 1 | Raltegravir, 2 | -5.23 | 9.88 | -0.53 | 0.96 |
|  | Dolutegravir, 2 | -14.47 | 5.80 | -2.49 | 0.12 |
| Raltegravir, 2 | Dolutegravir, 2 | -9.23 | 9.94 | -0.92 | 0.83 |

Raltegravir 1 –a group of participants on RAL baseline assessment; Raltegravir 2 –a group of participants on RAL second assessment; Dolutegravir 1 –a group of participants on DTG baseline assessment; Dolutegravir 2 –a group of participants on DTG second assessment

## Discussion and conclusions

This study aimed to follow up on neurocognitive performance in PWH on two types of INSTI-based ART at two time points, one year apart. We analysed this general goal through two specific ones to determine the neurocognitive functioning of PWH who are on INSTI-based regimen and to verify the existence of changes in neurocognitive functioning over time, depending on the type of integrase inhibitor they are taking (RAL/DTG) and the duration of therapy.

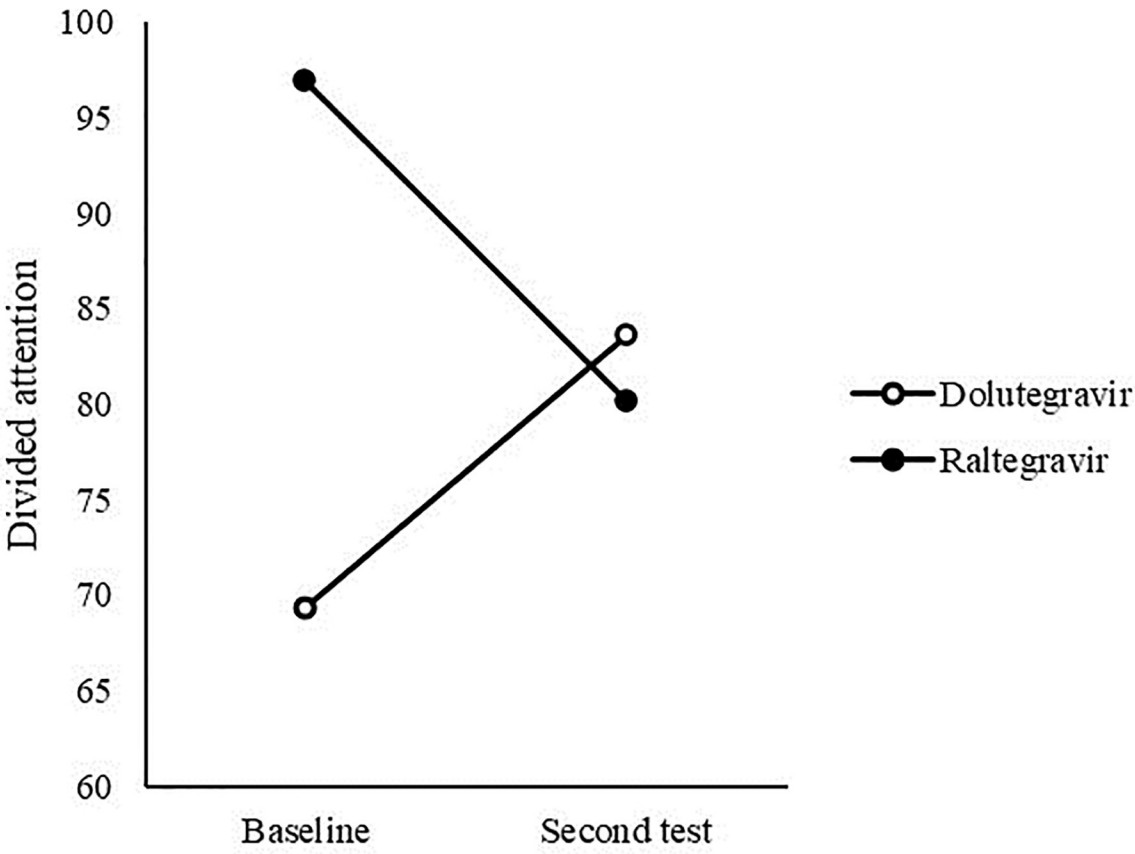

**Fig 1. Interactive effect of time and type of therapy on divided attention measure.**

We found no correlation of neurocognitive achievement with age, education level, backbone therapy and duration of treatment. The results also show no significant decrease in neurocognitive achievement on the MOCA, verbal working memory, cognitive capacity, sustained attention, divided attention, and verbal fluency measures over one-year follow-up. Also, no differences were observed in the effect of the therapy they are taking on any of the measures of neurocognitive functioning. A statistically significant interactive effect of time and type of therapy was found on the measure of divided attention, where we found that the DTG group slightly improved on the divided attention test, whereas the RAL group showed a slight decrease in performance on the same test. We also measured changes in the number of CD4 and CD8 cells and the inflammation parameter CD4/CD8 ratio. The results showed no decrease in the number of lymphocytes and the coefficient of inflammation, regardless of the type of ART the participants were taking.

In earlier research, it was shown that sociodemographic characteristics such as older age and, consequently, prolonged use of HIV therapy [23] may influence the likelihood of developing lower neurocognitive capacity [1]. In our research, this connection was not observed. However, the follow-up was limited to one year. These results support the idea that using INSTI-based regimen can make some cognitive improvements in PWH individuals [29]. Although the results from our study are in accordance with the previously mentioned optimistic perspective, we should be aware that the one-year follow-up period is not long enough to observe the cumulative effect of age and HIV-related neurodegeneration.

The fact that there are no significant differences in the neurocognitive functioning of PWH in any of the assessed measures during repeated measurements supports the assumption that with the ART regimen, the occurrence of HIV-associated neurocognitive disorders is less frequent and less intense [16, 17]. In our study, neurocognitive capacity remains stable on INSTI-based regimen.

RTG is the first HIV-integrase inhibitor approved by FDA for the treatment of HIV infection [14]. DTG is a second-generation INSTI with some advantages compared to other available antiretroviral agents [11–13], and we tried to find out whether there are differences in the effects of different types of INSTI-based regimen on the neurocognitive functioning of PWH. Our results indicate that both types of therapy are equally effective and keep a person's neurocognitive functioning stable. These results agree with other studies that also found that INSTI-based ART regimens were highly effective, with no significant differences between any of the INSTIs. In addition, side effects were rarely observed and were very mild [38].

When observing the interaction effect of the type of therapy and time point of assessment, we saw that it was statistically significant on the measure of divided attention, DTG group showed slight improvement, whereas the RTG group showed a slight decrease in performance.

These findings are in agreement with some studies that claim there is a trend towards improvement of neurocognitive function in HIV-positive treatment patients who receive three months of DTG-based ART [39]. At the same time, different studies warn of the harmful effects of DTG through an increase in body weight and consequent increase in peripheral inflammation and the occurrence of metabolic syndrome in PWH [40], and it is necessary to conduct additional studies with a multi-year follow-up.

We used the MoCA general score to measure overall cognitive performance as a screening tool for HAND among PWH. The MoCA also assesses several cognitive domains: short-term memory recall tasks, visuospatial abilities, attention, concentration and working memory, executive functions, language, abstract reasoning, and orientation. Therefore, a recommendation for future research would be to examine the differences in relation to those specific domains.

Some limitations of this study include the small sample size, especially patients with HAND, where all participants were males and without comorbidities. Also, the duration of

therapy in patients until baseline testing is not uniform. Although in the research, we statistically controlled the effect of the duration of the therapy, there is room for improving the study in relation. To obtain the most reliable results, the subjects included in the study were uniform in relation to their level of education. Also, all participants had good health status and HIV infection under control and undetectable virus via PCR test. By opting for a highly selected sample (in the sense of controlling all comorbidities that affect cognition and equalising sociodemographic characteristics), it is possible that we "lost" part of the information about the research problem because we did not include those PWH who have low education, with comorbid diseases and poor health status and condition.

This study was designed as a pilot proof of concept study because we have only a one-year follow-up, which is a relatively short period to make explicit assumptions about the effect of both types of INSTI on neurocognitive performance. We plan to conduct a longitudinal study with the same goal.

In this study, we also monitored the effects of the duration of therapy. Still, the limitation was that we did not assess participants' neurocognitive performance before beginning INSTI-based treatment and dosage of INSTI treatment. Although we have shown that with INSTI treatment, there is no significant worsening of neurocognitive status, the question remains whether INSTI drug regimens can protect or treat mild HIV-associated neurocognitive disorder.

Nonetheless, this is the first longitudinal study in Serbia aiming to explore HIV-associated neurocognitive impairment at more than one-time point and also the first study exploring the effects of duration and type of INSTI-based treatment.

Notwithstanding the limitations, our findings are consistent with recommendation that the second generation INSTI regimen has no or negligible influence on the neurocognitive functioning of PWH. Also, it is essential in theoretical and practical cross-cultural approaches to understand HIV infection and treatment in relation to the neurocognitive functioning of PWH, giving the basis to assume that both therapies from the INSTI group have the same effect on the neurocognitive functioning of PWH.

## Supporting information

**S1 File.**
(SAV)

## Author Contributions

**Conceptualization:** Nina Brkić-Jovanović, Snežana Brkić, Vojislava Bugarski-Ignjatović.

**Data curation:** Nina Brkić-Jovanović, Mina Karaman, Vojislava Bugarski-Ignjatović.

**Formal analysis:** Mina Karaman.

**Investigation:** Nina Brkić-Jovanović, Vanja Andrić, Snežana Brkić.

**Methodology:** Nina Brkić-Jovanović.

**Project administration:** Vanja Andrić, Vojislava Bugarski-Ignjatović.

**Resources:** Daniela Marić.

**Software:** Mina Karaman.

**Supervision:** Daniela Marić, Snežana Brkić.

**Validation:** Mina Karaman.

**Visualization:** Nina Brkić-Jovanović, Vanja Andrić, Snežana Brkić.

**Writing – original draft:** Nina Brkić-Jovanović, Vojislava Bugarski-Ignjatović.

**Writing – review & editing:** Daniela Marić.

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
