## [Decision Letter · Decision Letter 0]

30 Apr 2024

PONE-D-24-09779NEUROCOGNITIVE PROFILE IN HIV SUBJECTS ON INSTI-REGIMEN- ONE YEAR FOLLOW UP: IS THERE ROOM FOR OPTIMISM?PLOS ONE

Dear Dr. Brkic-Jovanovic,

Thank you for submitting your manuscript to PLOS ONE. After careful consideration, we feel that it has merit but does not fully meet PLOS ONE’s publication criteria as it currently stands. Therefore, we invite you to submit a revised version of the manuscript that addresses the points raised during the review process. Please make sure to address the requirements for extra details from the reviewers.

We look forward to receiving your revised manuscript.

Kind regards,

Lucette A Cysique, PhD

Academic Editor

PLOS ONE

Journal Requirements:

   "This work is supported in part by a research grant from the Investigator-Initiated Studies Program of Merck Sharp & Dohme doo."

5. For studies involving third-party data, we encourage authors to share any data specific to their analyses that they can legally distribute. PLOS recognizes, however, that authors may be using third-party data they do not have the rights to share. When third-party data cannot be publicly shared, authors must provide all information necessary for interested researchers to apply to gain access to the data. (https://journals.plos.org/plosone/s/data-availability#loc-acceptable-data-access-restrictions) 

7. Your ethics statement should only appear in the Methods section of your manuscript. If your ethics statement is written in any section besides the Methods, please move it to the Methods section and delete it from any other section. Please ensure that your ethics statement is included in your manuscript, as the ethics statement entered into the online submission form will not be published alongside your manuscript. 

Reviewers' comments:

Reviewer's Responses to Questions

**Comments to the Author**

1. Is the manuscript technically sound, and do the data support the conclusions?

Reviewer #1: Partly

Reviewer #2: Partly

2. Has the statistical analysis been performed appropriately and rigorously? 

Reviewer #1: Yes

Reviewer #2: Yes

3. Have the authors made all data underlying the findings in their manuscript fully available?

Reviewer #1: Yes

Reviewer #2: Yes

4. Is the manuscript presented in an intelligible fashion and written in standard English?

Reviewer #1: Yes

Reviewer #2: Yes

5. Review Comments to the Author

Reviewer #1: This is an interesting study. The major limitations are the short time frame for repeat testing, the small number of patients with HAND and the lack of any power analyses to address the issue of whether INSTI drug regimens can protect or treat mild HIV associated neurocognitive disorder

Reviewer #2: PONE-D-24-09779 compared the effects of RAL and DTG on neurocognition among newly diagnosed PWH during a one-year follow-up. The conclusion was no significant difference between these regimens. In general, the MS was well written with sound introduction, detailed statistical analyses, and discussion. My comments are brief on certain critical information that is missing in the MS.

1. The concurrent drug use including substances should be included. Certain medications and many substances have a direct impact on neurocognition. Thus, this information is critical.

2. The sample size calculation is missing. It is unclear how this sample size in each group was determined to detect the difference in cognition between the study groups. At least a discussion of power analysis should be provided.

3. A discussion of additional domain specific cognitive assessment to MoCA should be considered, particularly the rationales and benefits from those assessments.

4. The details of RAL and DTG are missing, including dosage, treatment interval and drug combinations.

6. PLOS authors have the option to publish the peer review history of their article (what does this mean?). If published, this will include your full peer review and any attached files.

Reviewer #1: No

Reviewer #2: **Yes: **Qing Ma

---

## [Author Response · Author response to Decision Letter 0]

14 May 2024

Response to Reviewers

Journal Requirements:

The complete work has now been prepared according to the instructions in the sent documents. Thanks for the suggestions.

Funding information has been removed from the manuscript.

Thanks for the correction; we have corrected the error.

 "This work is supported in part by a research grant from the Investigator-Initiated Studies Program of Merck Sharp & Dohme doo."

We have supplemented the statement in the requested manner.

5. For studies involving third-party data, we encourage authors to share any data specific to their analyses that they can legally distribute. PLOS recognizes, however, that authors may be using third-party data they do not have the rights to share. When third-party data cannot be publicly shared, authors must provide all information necessary for interested researchers to apply to gain access to the data. (https://journals.plos.org/plosone/s/data-availability#loc-acceptable-data-access-restrictions) 

I once again checked the issue of the possibility of sharing and having open data.

The results of the study of which this paper is a part exist site clinical trials. I am sending the registration statement as a document.

I asked our sponsors and received an answer: “The PI owns the data for this study as it is considered a non-Merck study with the institution as the sponsor. The collected data belong to the Faculty of Medicine research team in Novi Sad. If necessary, we can share the data with the protection of the respondents' personal data.

Answered in the previous section.

7. Your ethics statement should only appear in the Methods section of your manuscript. If your ethics statement is written in any section besides the Methods, please move it to the Methods section and delete it from any other section. Please ensure that your ethics statement is included in your manuscript, as the ethics statement entered into the online submission form will not be published alongside your manuscript. 

We have moved the ethics statement in the Methods section.

Comments to the Author

Reviewer #1: This is an interesting study. The major limitations are the short time frame for repeat testing, the small number of patients with HAND and the lack of any power analyses to address the issue of whether INSTI drug regimens can protect or treat mild HIV associated neurocognitive disorder

Thank you for the positive evaluation. We agree with the study's recognised limitations. We included and emphasised all of the above in the section dealing with the limitations of this research and recommendations for future research.

Reviewer #2: PONE-D-24-09779 compared the effects of RAL and DTG on neurocognition among newly diagnosed PWH during a one-year follow-up. The conclusion was no significant difference between these regimens. In general, the MS was well written with sound introduction, detailed statistical analyses, and discussion. My comments are brief on certain critical information that is missing in the MS.

Thank you for the positive evaluation.

1. The concurrent drug use including substances should be included. Certain medications and many substances have a direct impact on neurocognition. Thus, this information is critical.

Thanks for such a useful suggestion. We entered data on the uniformity of the group in the sense of not taking any additional therapy for diseases other than HIV. We also entered the data and checked the correlations with the backbone therapy that the subjects were taking. There were three types of backbone therapy in the study: tenofovir, abacavir and lamivudine, whose effect was controlled.

2. The sample size calculation is missing. It is unclear how this sample size in each group was determined to detect the difference in cognition between the study groups. At least a discussion of power analysis should be provided.

We included the sample size calculation and the fact that the sample size is one of this research's limitations.

3. A discussion of additional domain specific cognitive assessment to MoCA should be considered, particularly the rationales and benefits from those assessments.

Slazemo se da bi bilo zanimljivo razmatrati specificne domene MoCA testa. U ovom istraživanju smo ovaj test koristili kao trijažni test za pojavu HANDa a određene specifične domene pokrili smo dodatnim testovima u celosti (brojanje, verbal fluency...). Predlog za analizu specificnih domena MoCA testa smo naveli kao preporuku za buduca istrazivanja.

We agree that it would be interesting to consider specific domains of the MoCA test. In this research, we used this test as a triage test for the appearance of HAND, and we covered certain specific domains with additional tests in their entirety (counting, verbal fluency...). We have listed the proposal for analysing specific domains of the MoCA test as a recommendation for future research.

4. The details of RAL and DTG are missing, including dosage, treatment interval and drug combinations.

We included the length of therapy and backbone therapy as control variables in the manuscript.

We do not have data on patient doses, so we have added that to the section on the study's limitations.

Many thanks for the extremely useful suggestions for improving the work.

---

## [Decision Letter · Decision Letter 1]

15 Jun 2024

NEUROCOGNITIVE PROFILE IN HIV SUBJECTS ON INSTI-REGIMEN- ONE YEAR FOLLOW UP: IS THERE ROOM FOR OPTIMISM?

PONE-D-24-09779R1

Dear Dr. Brkic-Jovanovic,

We’re pleased to inform you that your manuscript has been judged scientifically suitable for publication and will be formally accepted for publication once it meets all outstanding technical requirements.

Kind regards,

Lucette A Cysique, PhD

Academic Editor

PLOS ONE

Reviewers' comments:

Reviewer's Responses to Questions

**Comments to the Author**

1. If the authors have adequately addressed your comments raised in a previous round of review and you feel that this manuscript is now acceptable for publication, you may indicate that here to bypass the “Comments to the Author” section, enter your conflict of interest statement in the “Confidential to Editor” section, and submit your "Accept" recommendation.

Reviewer #1: All comments have been addressed

Reviewer #2: All comments have been addressed

2. Is the manuscript technically sound, and do the data support the conclusions?

Reviewer #1: Yes

Reviewer #2: Yes

3. Has the statistical analysis been performed appropriately and rigorously? 

Reviewer #1: Yes

Reviewer #2: Yes

4. Have the authors made all data underlying the findings in their manuscript fully available?

Reviewer #1: Yes

Reviewer #2: Yes

5. Is the manuscript presented in an intelligible fashion and written in standard English?

Reviewer #1: Yes

Reviewer #2: Yes

6. Review Comments to the Author

Reviewer #1: The authors have adequately addressed the previous issues raised. Therefore I have no further comments

Reviewer #2: Thanks for addressing my comments. The missing information such as dosage did present some challenges, but the discussion of limitations was sufficient for this MS. Also the plan for domain specific analysis was sound. Thanks

7. PLOS authors have the option to publish the peer review history of their article (what does this mean?). If published, this will include your full peer review and any attached files.

Reviewer #1: No

Reviewer #2: **Yes: **Qing Ma

---

## [Editor Report · Acceptance letter]

18 Jun 2024

PONE-D-24-09779R1 

PLOS ONE

Dear Dr. Brkic-Jovanovic, 

I'm pleased to inform you that your manuscript has been deemed suitable for publication in PLOS ONE. Congratulations! Your manuscript is now being handed over to our production team.

Kind regards, 

on behalf of

Dr. Lucette A Cysique 

Academic Editor

PLOS ONE